# Long- versus short-duration systemic corticosteroid regimens for acute exacerbations of COPD: A systematic review and meta-analysis of randomized trials and cohort studies

**Zhen Zhao**[1]*, **Owen Lou**[2], **Yiyang Wang**[3], **Raymond Yin**[1], **Carrie Gong**[4], **Florence Deng**[2], **Ethan C. Wu**[5], **Jing Yi Xie**[2], **Jerry Wu**[3], **Avery Ma**[6], **Yongzhi Guo**[7], **Wei Ting Xiong**[7]

1 Western University, London, Ontario, Canada, 2 McMaster University, Hamilton, Ontario, Canada, 3 University of California, Los Angeles, Los Angeles, California, United States of America, 4 Bayview Secondary School, Richmond Hill, Ontario, Canada, 5 Sir Winston Churchill High School, Calgary, Alberta, Canada, 6 Collège Charlemagne, Pierrefonds, Québec, Canada, 7 University of Toronto, Toronto, Ontario, Canada

* markzhao2@gmail.com

## Abstract

While systemic corticosteroids quicken patient recovery during acute exacerbations of COPD, they also have many adverse effects. The optimal duration of corticosteroid administration remains uncertain. We performed a systematic review and meta-analysis to compare patient outcomes between short- (≤7 days) and long- (>7 days) corticosteroid regimens in adults with acute exacerbations of COPD. MEDLINE, EMBASE, CENTRAL, and hand searches were used to identify eligible studies. Risk of bias was assessed using the Cochrane RoB 2.0 tool and ROBINS-I. Data were summarized as ORs (odds ratios) or MDs (mean differences) whenever possible and qualitatively described otherwise. A total of 11532 participants from eight RCTs and three retrospective cohort studies were included, with 1296 from seven RCTs and two cohort studies eligible for meta-analyses. Heterogeneity was present in the methodology and settings of the studies. The OR (using short duration as the treatment arm) for mortality was 0.76 (95% CI = 0.40–1.44, n = 1055). The MD for hospital length-of-stay was -0.91 days (95% CI = -1.81–-0.02 days, n = 421). The OR for re-exacerbations was 1.31 (95% CI = 0.90–1.90, n = 552). The OR for hyperglycemia was 0.90 (95% CI = 0.60–1.33, n = 423). The OR for infection incidence was 0.96 (95% CI = 0.59–1.156, n = 389). The MD for one-second forced expiratory volume change was -18.40 mL (95% CI = -111.80–75.01 mL, n = 161). The RCTs generally had low or unclear risks of bias, while the cohort studies had serious or moderate risks of bias. Our meta-analyses were affected by imprecision due to insufficient data. Some heterogeneity was present in the results, suggesting population, setting, and treatment details are potential prognostic factors. Our evidence suggests that short-duration treatments are not worse than long-duration treatments in moderate/severe exacerbations and may lead to considerably better outcomes in milder exacerbations. This supports the current GOLD guidelines.

**Data Availability Statement:** All relevant data are within the manuscript and its Supporting Information files.

**Funding:** The authors received no specific funding for this work.

**Competing interests:** The authors have declared that no competing interests exist.

**Trial registration:** Our protocol is registered in PROSPERO: CRD42023374410.

## Introduction

Chronic obstructive pulmonary disease (COPD) is a leading cause of mortality and morbidity worldwide, causing 3.3 million deaths and 71.1 million disability-adjusted life years in 2019, primarily in lower-income countries [1]. It is a generally irreversible condition, with the largest risk factor being tobacco smoking, although biomass burning, occupational hazards, and air pollution can also play a considerable role [2]. While COPD is chronic, many patients also suffer from acute exacerbations of symptoms which can last from days to weeks [3]. Respiratory infection is a major contributor to exacerbation, with others including environmental irritants such as pollutants and temperature [4]. While milder exacerbations can often be managed in an outpatient setting, severe exacerbations often require hospitalization and represent a major source of mortality and economic cost [5–7]. Furthermore, frequent exacerbations are correlated with faster deterioration of lung function, though it is unclear if this is a causal relationship [8].

COPD is usually diagnosed via spirometry using an individual's post-bronchodilator one-second forced expiratory volume ($FEV_1$) and forced vital capacity (FVC). According to the GOLD guidelines, an $FEV_1$/FVC <0.7 is necessary for COPD diagnosis [9]. The severity of airflow limitation is assessed via an individual's % predicted $FEV_1$, which is a proportion of what would be expected from a healthy person. These values are clinically important as they affect the recommended selection of treatments by the physician [9]. However, the ERJ/ATS and some researchers favour the alternative use of z-scores to define a "lower limit of normal" (LLN) of lung function [10–12], with an LLN of z = -1.64 (5th percentile) being pathological. Unlike GOLD's ratio-based method, z-scores account for confounding factors such as age and sex [13, 14]. Despite this, the 2023 GOLD guidelines continue to favour ratios due to their simplicity and the fact that consideration of the broader clinical context will mitigate the risk of misdiagnosis [15]. In any case, recent clinical trials have continued to use the GOLD method to define COPD and its severity [16–19].

For mild exacerbations, GOLD guidelines recommend treatment using only short-acting bronchodilators, whereas for moderate and severe exacerbations corticosteroid administration should also be considered [9]. Corticosteroids have anti-inflammatory properties, which are useful since COPD exacerbations are strongly associated with both local and systemic inflammation [20]. There is strong evidence that systemic corticosteroids improve patient outcomes during moderate and severe exacerbations [21, 22]. However, there are also many adverse effects associated with short-term corticosteroid use, such as hyperglycemia, gastrointestinal complications, and infection due to immunosuppression [20, 23]. These effects are directly correlated with the dosage and duration of corticosteroid administered [20, 23]. Thus, they are not usually recommended as maintenance therapy for stable COPD and there is interest in minimizing the amount of corticosteroid given during exacerbations while still retaining their benefits [9]. The recommended duration of corticosteroid regimens has decreased from less than two weeks to less than one week in the recent decades [24, 25].

There is no strong consensus on the duration of corticosteroids that should be given to a patient during exacerbations, with studies hampered by a seemingly large degree of patient-to-patient variation in their responses to corticosteroid treatment [20]. Although guidelines from GOLD and other organizations exist, poor adherence during exacerbations remains an issue

and is related to worse outcomes for patients [26, 27]. A systematic review and meta-analysis conducted by Walters et al. in 2018 found that short-duration systemic corticosteroid regimens ($\leq$7 days) are not likely to lead to worse outcomes than long-duration regimens ($>$ 7 days) in adult patients with COPD exacerbations [28]. However, the authors also concluded that there were not enough data to form a definitive conclusion, especially for patients with mild or moderate COPD. In this study, we revisited the comparison with additional data from a large RCT and three retrospective cohort studies—one of which specifically examined outpatients—which were not included in the 2018 review. These data increased our confidence that short-duration regimens are not inferior in terms of mortality, re-exacerbation, infection, hyperglycemia, hospital length-of-stay (LOS), and $FEV_1$ change.

## Methods

This systematic review and meta-analysis was conducted in accordance with the PRISMA 2020 guidelines and the protocol registered in PROSPERO (ID: CRD42023374410) [29]. Our completed PRISMA 2020 checklist is available in S1 Fig.

### Information sources and search strategy

To find studies, we systematically searched MEDLINE, EMBASE, and CENTRAL using Ovid. Medical Subject Heading terms were used to define the search strategy, which is available in S2 Fig. No language or date restrictions were imposed on the searches, which were conducted in September 2022. Additionally, references from similar previous systematic reviews were hand-searched in May 2023 to identify studies missed by the database search.

### Selection process

After removing duplicates, studies were imported into Covidence—a web-based screening platform by Cochrane—where they were each subjected to a round of title/abstract screening followed by a round of full-text screening [30]. In each round, two reviewers independently assessed each study, and disagreements were resolved through discussion involving a third reviewer. Each study required unanimous approval from its reviewers to advance into the next stage.

Studies were included if they included adults (18 years or older) who were receiving corticosteroids as an acute-phase treatment for an exacerbation of COPD as defined by each individual study. Each study had to include a group who received a short-duration corticosteroid regimen ($\leq$7 days) and a group who received a long-duration corticosteroid regimen ($>$7 days). All corticosteroid types and administration methods were included, as well as studies with co-interventions such as bronchodilators and antibiotics. Studies using corticosteroids for maintenance therapy (as opposed to treatment for exacerbations) and studies including asthmatic patients were excluded. While we originally intended to also examine the differences between low and high doses of corticosteroids in our protocol, there was too much heterogeneity among the studies during the preliminary screening. Thus, our review only examined the differences between short- and long-duration regimens.

### Data extraction

Each study was randomly assigned to two reviewers who independently extracted data, with conflicts being resolved through discussion and arbitration by a third reviewer. The data extraction form was created *a priori* and included the outcomes mortality, length of hospital stay, number of re-exacerbations during follow-up, $FEV_1$ change, respiratory infection, and hyperglycemia. All data points within the follow-up period were included. Additional collected information included

bibliographic information, study type (RCT or cohort), details of corticosteroid regimen, study period, country, definition of exacerbation, inclusion/exclusion criteria, co-interventions, comorbidities, time until follow-up starts, follow-up duration, and demographic data.

Extraction for Alshehri 2021, Chen 2008, Leuppi 2013, and Gomaa 2008 were done by OL and RY; Al Mamun 2011, Sayiner 2001, Zhou 2021, and Poon 2020 by ZZ and FD; Wood-Baker 1997, Sirichana 2008, and Sivapalan 2019 by JYX and YW; and WTX performed arbitration.

### Risk of bias assessment

Risk of bias assessment for each study was conducted independently and in duplicate by the same reviewers who performed the data extraction for that study. RCTs were assessed using the Cochrane RoB 2.0 tool which assessed five domains: bias due to the randomization process, deviations from intended interventions, missing outcome data, measurement of outcome, and selection of the reported result [31]. Cohort studies were assessed using the ROBINS-I tool which assessed seven domains: bias due to confounding, selection of patients into the study, classification of intervention, deviations from intended intervention, missing data, measurement of outcomes, and selection of reported results [32]. We tried to minimize the risk of publication bias by using a broad search strategy. We also intended to use funnel plots and statistical testing to assess the risk of publication bias but were unable to do so due to an insufficient number of studies.

### Data analysis

Data from studies were compiled into meta-analyses and visualized where possible using the RevMan Web application (version 5.8.0) from Cochrane [33]. Mantel-Haenszel odds ratios (ORs) were used for the number of re-exacerbation, hyperglycemia, and infection events during follow-up, while Peto ORs were used for mortality due to the rarity of events. Mean differences (MDs) were used for hospital length-of-stay and FEV change. Point estimates were presented along with the 95% confidence interval (CI) and p-value for the combined data and RCT-only data for each outcome, along with cohort study-only data if it was significant. For each meta-analysis, heterogeneity was assessed using visual inspection of the forest plots as well as the $I^2$ statistic. $I^2$ values greater than 50% were considered significantly heterogeneous while values greater than 75% were considered seriously heterogeneous. Fixed effects models were used for analyses, with sensitivity analyses using random effects models being conducted when significant unexplainable heterogeneity was present.

We attempted to contact study authors to obtain missing data. If needed, missing standard deviations were estimated using methods described in the Cochrane Handbook for Systematic Reviews. Certainty of evidence was assessed using the GRADE (Grading of Recommendations, Assessment, Development, and Evaluations) framework, with each outcome being graded as being of high, moderate, low, or very low certainty [34]. In accordance with guidelines, outcomes started off as "high" certainty, and the ratings were lowered by one level for a moderate degree (or two levels for a serious degree) of risk-of-bias, imprecision, inconsistency, indirectness, or publication bias. Two authors independently assessed each outcome, with any disagreements being resolved through discussion.

## Results

### Search results

After removing duplicates, 4947 studies were identified from database and hand searches (Fig 1). Eight RCTs and three retrospective cohort studies were included in our review, with

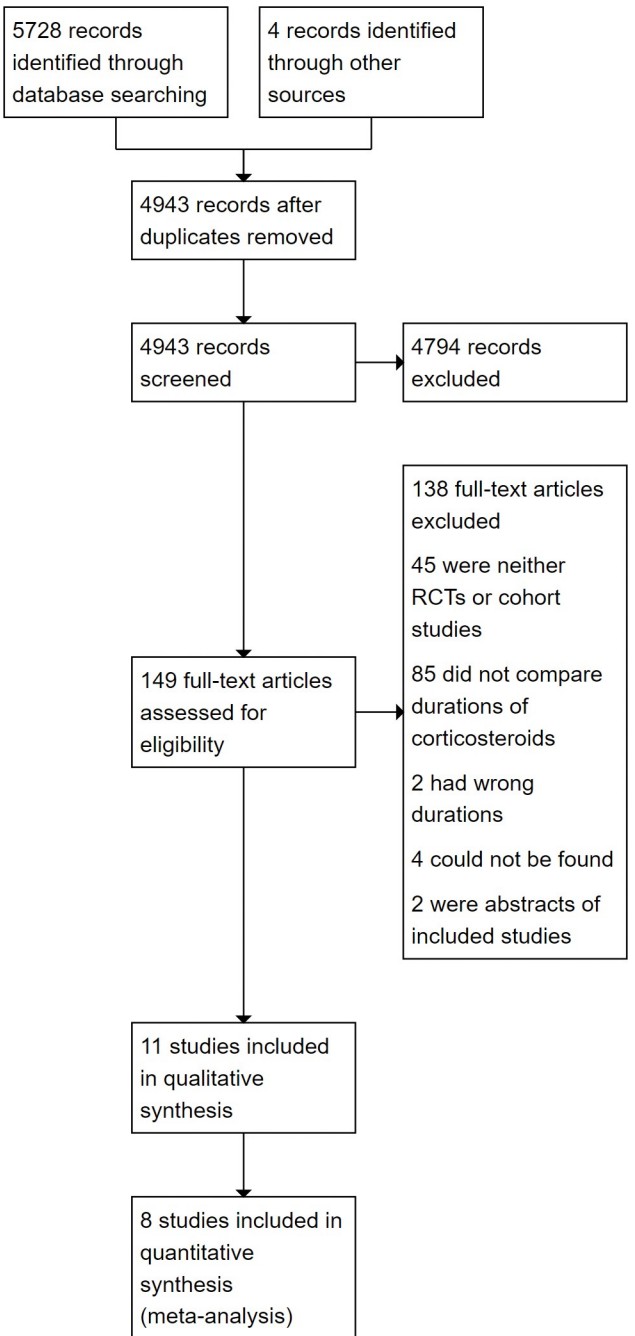

**Fig 1. PRISMA flow chart outlining the search and selection process.**

seven RCTs and two cohort studies being included in the meta-analyses. The characteristics of the studies are summarized in Table 1. Of the eight RCTs, four were only available as abstracts. We also referenced the 2018 Cochrane review by Walters et al. to fill gaps in our data [28]. There were four records in our search for which we could not find the article for, but their titles suggested none of them would have fit our inclusion criteria. Excluded studies which were similar to those we included are listed in S1 Table along with the reason for exclusion. The main results from the included studies are summarized in Table 2.

**Table 1. Summary of important characteristics of studies included in the review.**

| Study ID | Definition of exacerbation | Definition of COPD | Inclusion/exclusion criteria | Interventions/ Groups | Study size | % of males | Age[a] |
|---|---|---|---|---|---|---|---|
| Al Mamun 2011 (RCT abstract) [35] | N/A | N/A | Inclusion: $FEV_1$ <50% predicted | 30 mg/day of oral prednisolone for 7 days | N/A | Not significantly different | N/A |
| | | | | 30 mg/day of oral prednisolone for 14 days | N/A | Not significantly different | N/A |
| Alshehri 2021 (Retrospective cohort) [36] | Diagnosis based on GOLD criteria | Diagnosis based on GOLD criteria | Inclusion: Primary diagnosis of acute COPD exacerbation and hospital admission a case of COPD exacerbation, and >40 years of age Exclusion: History of asthma, pneumonia, prior home use of systemic corticosteroids, or unavailability of treatment plan | Systemic corticosteroids for ≤5 days | 28 | 64.3% | 68.3 ± 1.3 |
| | | | | Systemic corticosteroids for >5 days | 52 | 55.8% | 66.2 ± 1.6 |
| Chen 2008 (RCT) [18] | Diagnosis based on GOLD criteria | $FEV_1/FVC < 0.7$ | Inclusion: Coughing with phlegm for >2 years; and $FEV_1$ <80% predicted; and at least two of: increased dyspnea, sputum quantity, or sputum purulence Exclusion: Respiratory failure, diabetes, or bronchial asthma | 30 mg/day of oral prednisone for 7 days, then 7 days placebo. | 41 | 72.7% | 70 ± 8 |
| | | | | 30 mg/day of oral prednisone for 14 days. | 40 | 79.1% | 72 ± 7 |
| Gomaa 2008 (RCT abstract) [37] | N/A | N/A | Inclusion: $FEV_1$<50% predicted Exclusion: Respiratory acidosis | 30 mg/day oral prednisolone for 7 days, unknown if placebo was given | Unknown (42 total) | N/A | N/A |
| | | | | 30 mg/day oral prednisolone for 14 days | Unknown (42 total) | N/A | N/A |
| Leuppi 2013 (RCT) [19] | At least 2 of the following: change in baseline dyspnea, cough, or sputum quantity, or sputum purulence | $FEV_1/FVC < 0.7$ | Inclusion: >40 years of age and smoking >20 pack-years. Exclusion: History of asthma, pneumonia, estimated survival < 6 months pregnancy or lactation, or inability to give informed consent | 40 mg of IV methylprednisolone on day 1, then 40 mg/day of oral prednisolone for 4 days, then 9 days placebo. | 156 | 67.3%[b] | 69.8 ± 11.3 |
| | | | | 40 mg of IV methylprednisolone on day 1, then 40 mg/day of oral prednisolone for 13 days. | 155 | 53.5%[b] | 69.8 ± 10.6 |
| Poon 2020 (Retrospective cohort) [38] | N/A | N/A | Inclusion: >18 years of age and given at least 1 day of high-dose IV methylprednisolone, admitted to ICU Exclusion: Readmission within 6 months of previous exacerbation, corticosteroids prematurely discontinued, or death during treatment | ≥40 mg q6h but <240 mg/day IV methylprednisolone for 6.9 ± 1.7 days, including a 4.3 ± 1.3 day taper period | 39 | 53.9% | Median/ IQR: 66 (58–79) |
| | | | | ≥40 mg q6h but <240 mg/day IV methylprednisolone for 16.5 ± 7.1 days, including a 13.9 ± 6.9 day taper period | 39 | 41.0% | Median/ IQR: 62 (58–82) |
| Sayiner 2001 (RCT) [39] | N/A | N/A | Inclusion: Exacerbation leading to hospitalization, smoking history ≥20 pack-years and $FEV_1$ <35% predicted, informed consent, severe dyspnea with sleeping difficulties, and respiratory failure Exclusion: Personal or family history of asthma, atopy, allergic disease, eosinophilia, use of systemic steroids in the preceding month, severe hypertension, uncompensated congestive heart failure, difficult-to-control diabetes mellitus, or mechanical ventilation | 0.5 mg/kg IV methylprednisolone q6h for 3 days, then placebo IV twice daily for 3 days, then placebo IV once daily for 4 days | 17 | 94.1% | 67.4 ± 5.8 |
| | | | | 0.5 mg/kg IV methylprednisolone q6h for 3 days, then 0.5 mg/kg q12h for 3 days, then 0.5 mg/kg/day for 4 days | 17 | 94.1% | 64.1 ± 9.1 |

(*Continued*)

**Table 1.** (Continued)

| Study ID | Definition of exacerbation | Definition of COPD | Inclusion/exclusion criteria | Interventions/ Groups | Study size | % of males | Age[a] |
|---|---|---|---|---|---|---|---|
| Sirichana 2008 (RCT abstract) [40] | At least 2 of three of the following: increased dyspnea, sputum quantity, or sputum purulence for at least 24 hours | N/A | Inclusion: >40 years of age | 30 mg/day prednisolone for 5 days, unknown if placebo was given | 24 | N/A | N/A |
| | | | | 30 mg/day prednisolone for 10 days | 22 | N/A | N/A |
| Sivapalan 2019 (Retrospective cohort) [16] | N/A | N/A | Inclusion: Registered in the Danish Register of COPD and received a prednisolone prescription for treatment of COPD exacerbation in an outpatient clinic in the Danish National Health Service Prescription Database<br>Exclusion: Ever had asthma diagnosis or prednisolone prescription was >2500 mg (commonly used for maintenance therapy) | Prednisolone prescription ≤250 mg (corresponding to a 5-day regimen) | 6002 | 49.4% | Median/ IQR: 70 (62–76) |
| | | | | Prednisolone prescription >250 mg (corresponding to a 10-day regimen) | 4150 | 51.4% | Median/ IQR: 70 (63–77) |
| Wood-Baker 1997 (RCT abstract) [41] | N/A | N/A | Inclusion: >40 years of age, >10 pack-year smoking history, and $FEV_1$ < 50% of predicted value<br>Exclusion: Corticosteroids for maintenance therapy, presence of lung diseases including pneumonia, previous adverse reaction to corticosteroids, peptic ulcer disease within the past 2 years, history of cardiac failure, hepatic or renal failure, or inadequately treated hypertension | 2.5mg/kg prednisolone for 3 days followed by 11 days placebo | 27 in total (including placebo arm) | 64.3% in total (including placebo arm) | Median/ IQR: 72 (61–86) in total (including placebo arm) |
| | | | | 0.6 mg/kg prednisolone for 7 days followed by 0.3 mg/kg for 7 days | 27 in total (including placebo arm) | 64.3% in total (including placebo arm) | Median/ IQR: 72 (61–86) in total (including placebo arm) |
| Zhou 2021 (RCT) [17] | Sudden changes in clinical symptoms, including dyspnea, coughing, and sputum production | $FEV_1$/ FVC < 0.7 | Inclusion: 40–70 years of age, COPD beyond the range of daily variation<br>Exclusion: Assisted respiratory muscle movement, contradictory breathing, cyanosis, need for invasive mechanical ventilation, edema, right heart failure, hemodynamic instability, changes in mental state, malignant tumours, or other serious illnesses | 40 mg/day IV methylprednisolone for 5 days, then saline placebo for 4 days (same amount as long-duration group) | 329 | 53.8% | 61.19 ± 5.1 |
| | | | | 40 mg/day IV methylprednisolone for 5 days, then 30 mg/day for 2 days, then 20 mg/day for 2 days | 310 | 55.5% | 60.87 ± 4.8 |

[a]Data presented as means and SDs unless otherwise specified.

[b]Proportion of males was significantly different (p = 0.02).

## Risk of bias

The Cochrane Risk of Bias 2.0 tool was used to assess the RCTs in our study. Three of the RCTs available as full articles (Leuppi 2013, Sayiner 2001, and Chen 2008) were deemed to be at low risk of bias. The fourth full article, Zhou 2021, was an open-label trial and rated as having moderate risk of bias. However, we lowered its risk to "low" for mortality and re-exacerbations due to the objectivity of the outcomes; a lack of blinding would not likely have affected the outcome measurement. The four abstract-only studies did not have enough information to

**Table 2. Summary of results for each outcome.**

| Outcome | Sample size | Summary estimate [95% CI] | Other data not included in meta-analyses | Certainty of evidence (GRADE) |
|---|---|---|---|---|
| Mortality | 1055 | OR 0.76 [0.40–1.44], p = 0.39 | Sivapalan 2019: aHR 1.8 [1.5–2.2] for outpatients (p<0.0001) | Moderate (⊕⊕⊕○) |
| Hospital LOS | 421 | MD -0.91 days [-1.81– -0.02 days], p = 0.05 | Alshehri 2021: p = 0.88<br>Poon 2020: 7 vs 11 days, p<0.05 | Moderate (⊕⊕⊕○) |
| Re-exacerbation | 552 | OR 1.31 [0.90–1.90], p = 0.16 | Zhou 2021: p>0.05 | Moderate (⊕⊕⊕○) |
| Hyperglycemia | 423 | OR 0.90 [0.60–1.33], p = 0.58 | Sirichana 2008: No difference in fasting plasma glucose | Moderate (⊕⊕⊕○) |
| Infection | 389 | OR 0.96 [0.59–1.56], p = 0.87 | Sivapalan 2019: aHR 1.2 [1.0–1.3] for pneumonia in outpatients (p = 0.011) | Moderate (⊕⊕⊕○) |
| $FEV_1$ Change | 161 | MD -18.40 mL [-111.80–75.01 mL], p = 0.70 | Leuppi 2013: No difference at discharge or 0, 6, 30, or 180 days (p = 0.94)<br>Zhou 2021: No difference in % of predicted $FEV_1$ after 180 days (p = 0.134).<br>Gomaa 2008: No difference after 7, 14, or 30 days<br>Al Mamun 2011: No difference after 7 or 14 days | Low (⊕⊕○○) |

properly assess overall risk of bias (Table 3). One abstract, Sirichana 2008, was an open-label trial and therefore was rated as "moderate" for bias in the "deviation from intended domains" and "measurement of the outcome" domains.

The ROBINS-I tool was used to assess risk of bias for the cohort studies by comparing their designs to hypothetical, well-designed RCTs assessing the same question in the same setting (Table 4). All three studies had issues with confounding and selection as the choice of corticosteroid regimen duration could have been affected by the severity of the patient's exacerbation. This was mitigated in Sivapalan 2019 as they specifically used data which spanned a guideline change for corticosteroid regimen duration in Denmark. This meant the choice between a shorter- and longer-duration regimen was more likely to be due to the specific guideline in effect at that time than the severity of exacerbation [16]. The studies were also rated as "moderate" for "risk of bias from selection of the reported result" as they did not report a separate protocol or statistical analysis plan. Alshehri 2021 was rated as "unknown" for "bias in deviations from the intended interventions" as it did not report data for some co-interventions and co-morbidities.

## Results of syntheses

**Mortality.** Three RCTs and one cohort study contributed a total of 1055 participants to the meta-analysis for mortality (Fig 2). One RCT (Wood-Baker 1997) reported zero deaths

**Table 3. Risk of bias assessment of included RCTs conducted using Cochrane RoB 2.0.**

| | Randomization Process | Deviations from intended interventions | Missing outcome data | Measurement of the outcome | Selection of the reported result | Overall bias |
|---|---|---|---|---|---|---|
| Al Mamun 2011 | Unknown | Unknown | Unknown | Unknown | Unknown | Unknown |
| Chen 2008 | Low | Low | Low | Low | Low | Low |
| Gomaa 2008 | Unknown | Unknown | Unknown | Unknown | Unknown | Unknown |
| Leuppi 2013 | Low | Low | Low | Low | Low | Low |
| Sayiner 2001 | Low | Low | Low | Low | Low | Low |
| Sirichana 2008 | Unknown | Moderate | Unknown | Moderate | Unknown | Unknown |
| Wood-Baker 1997 | Unknown | Unknown | Unknown | Unknown | Unknown | Unknown |
| Zhou 2021 | Low | Low | Low | Moderate | Low | Low |

**Table 4. Risk of bias assessment of included cohort studies conducted using ROBINS-I.**

| | Bias due to Confounding | Bias in selection of participants into the study | Bias in classification of interventions | Bias in deviations from the intended interventions | Bias due to missing data | Bias in measuring of outcomes | Bias in selection of the reported result | Overall bias |
|---|---|---|---|---|---|---|---|---|
| Alshehri 2021 | Serious | Serious | Low | No Info | Low | Low | Moderate | Serious |
| Poon 2020 | Serious | Serious | Low | Low | Low | Low | Moderate | Serious |
| Sivapalan 2019 | Moderate | Moderate | Low | Low | Low | Low | Moderate | Moderate |

and was unable to produce an OR. The combined OR between the long- and short-duration groups was 0.76 (95% CI = 0.40–1.44, p = 0.39), while the OR with only RCT data was 0.87 (95% CI = 0.44–1.70, p = 0.68).

In contrast, another cohort study by Sivapalan et al. found that corticosteroid regimens over 10 days resulted in higher mortality compared to those less than 10 days, with an adjusted hazard ratio of 1.8 (95% CI = 1.5–2.2) one year after treatment. Notably, this study was conducted using a Danish outpatient registry, whereas the other studies were conducted on inpatients with comparatively more severe exacerbations.

Both Leuppi 2013 and Zhou 2021 were judged to be at low risk of bias for this outcome. Alshehri 2021 had serious risk of bias, while Sivapalan 2019 had moderate risk of bias. The certainty of evidence from the RCT data was downgraded to "moderate" due to imprecision, while the certainty from cohort study data was downgraded to "very low" due to imprecision and risk of bias. Altogether, our data shows there is likely no difference in mortality between

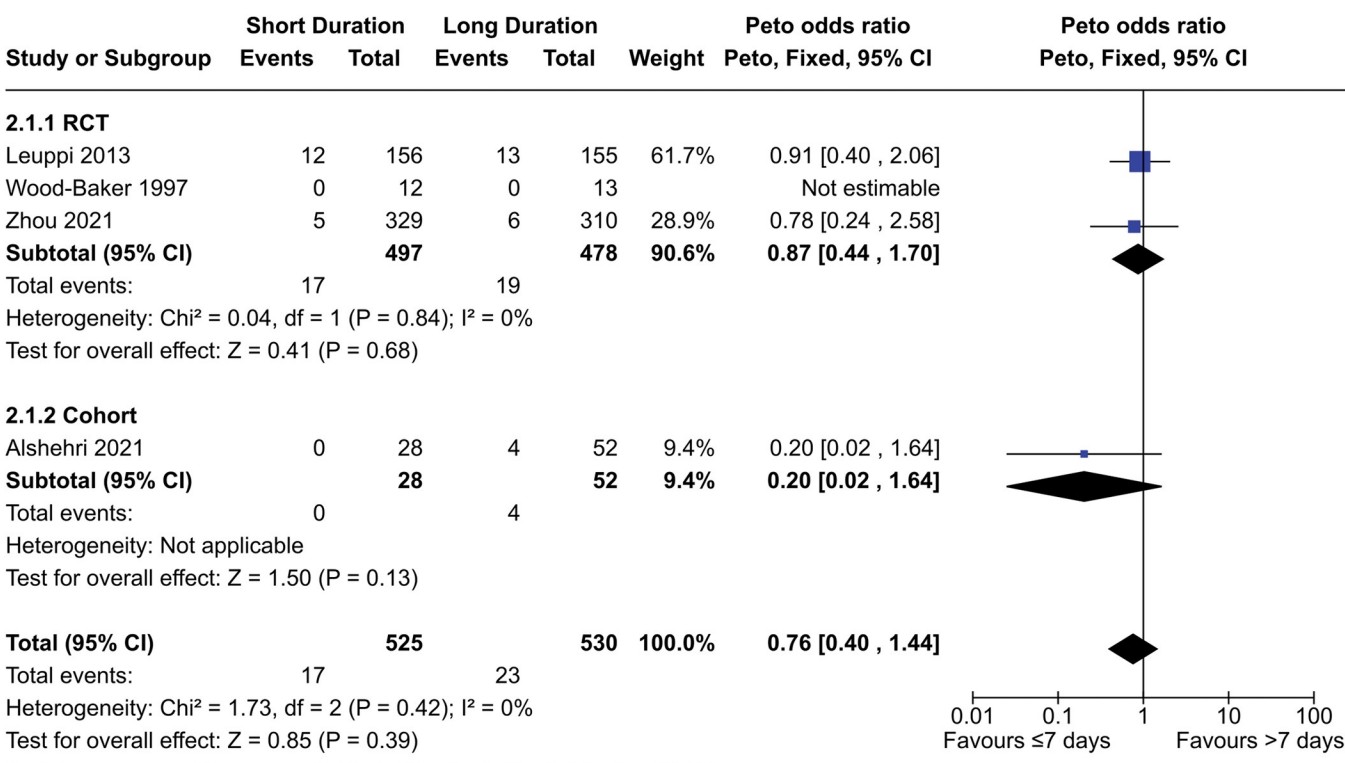

**Fig 2. Comparison of mortality risk between short- (≤7 days) and long- (>7 days) duration regimens of corticosteroids.**

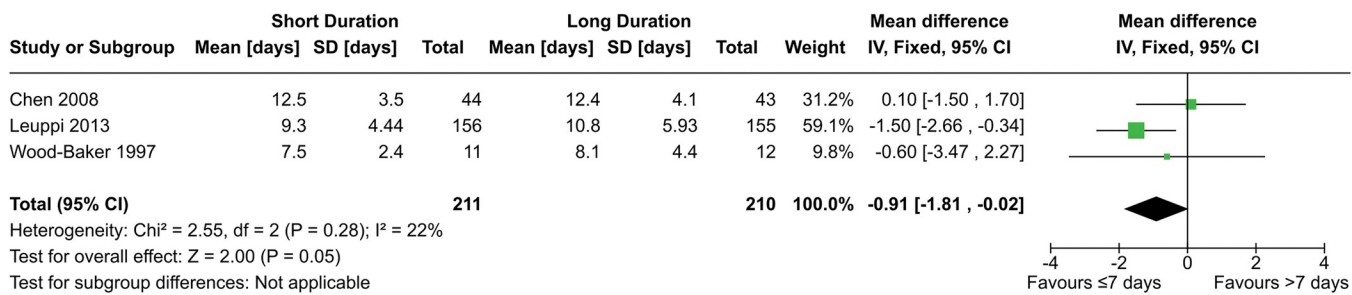

**Fig 3. Comparison of hospital length-of-stay between short- (≤7 days) and long- (>7 days) duration regimens of corticosteroids.**

short- (≤7 days) and long- (>7 days) duration regimens, though mortality may be lower in short-duration regimens specifically in outpatients with less-severe exacerbations.

**Hospital length-of-stay.** Three RCTs contributed a total of 421 participants to the meta-analysis for hospital length-of-stay (Fig 3). One of the RCTs (Leuppi 2013) did not report SDs, instead only reporting medians, IQR, and means. We used the reported means and estimated the SDs by dividing the IQRs by 1.35. However, this method relies on the assumption that the data is normally distributed, so we performed a sensitivity analysis by excluding the study. With Leuppi 2013 included, the mean difference is marginally significant at -0.91 days (95% CI = -1.81–-0.02 days, p = 0.05). However, excluding the study causes the difference to become non-significant at -0.07 days (95% CI = -1.47–1.33 days, p = 0.93). Alternative methods of interpretation yield different results; a similar analysis conducted in a previous review by Walters et al. used the medians instead of the means and produced a non-significant result of -0.61 days (95% CI = -1.51–0.28 days, p = 0.18) [28]. Notably, the significance of the difference in length of stay was reported as p = 0.04 in Leuppi 2013 via the log-rank test, which is likely closer to the null than the data in our meta-analysis [19]. The true degree of significance is unknown, but we believe it is unlikely to be strongly significant.

Two cohort studies were unable to be included in the meta-analysis as they did not report the required statistics. Alshehri 2021 found no significant difference (p = 0.88) between long-duration and short-duration regimens, while Poon 2020 found shorter stays in the short regimen group (p<0.05).

Chen 2008 and Leuppi 2013 had low risk of bias, while Wood-Baker 1997 had unknown risk. Alshehri 2021 and Poon 2020 had serious risk of bias. The evidence from the RCTs was downgraded to "moderate" due to imprecision, while the evidence from the cohort studies were downgraded to "very low" due to risk of bias and imprecision. Altogether, our data shows that hospital LOS is likely to be either the same between short- (≤7 days) and long- (>7 days) duration regimens or slightly shorter in short-term regimens.

**Number of re-exacerbations.** Four RCTs and one cohort study contributed a total of 552 participants towards the meta-analysis for the number of re-exacerbations during follow-up (Fig 4). The odds ratio for all the data was 1.31 (95% CI = 0.90–1.90, p = 0.16), while the ratio for RCT data only was 1.04 (95% CI = 0.70–1.56, p = 0.84). The odds ratio for the cohort study was significant at 5.21 (95% CI = 1.93–14.08, p = 0.001). There was considerable heterogeneity in the data ($I^2$ = 56%) due to the cohort study. Bias notwithstanding, it is possible this difference is due to the different study population; the RCTs were conducted in Europe and east Asia, while Alshehri 2021 used data from Saudi Arabia. Alshehri et al. noted the low compliance rate for COPD maintenance therapy in the Middle East, with low compliance being linked to higher exacerbation rates [36]. However, the impact resulting from this difference in populations is unclear. Another difference is follow-up duration, with Alshehri 2021 and

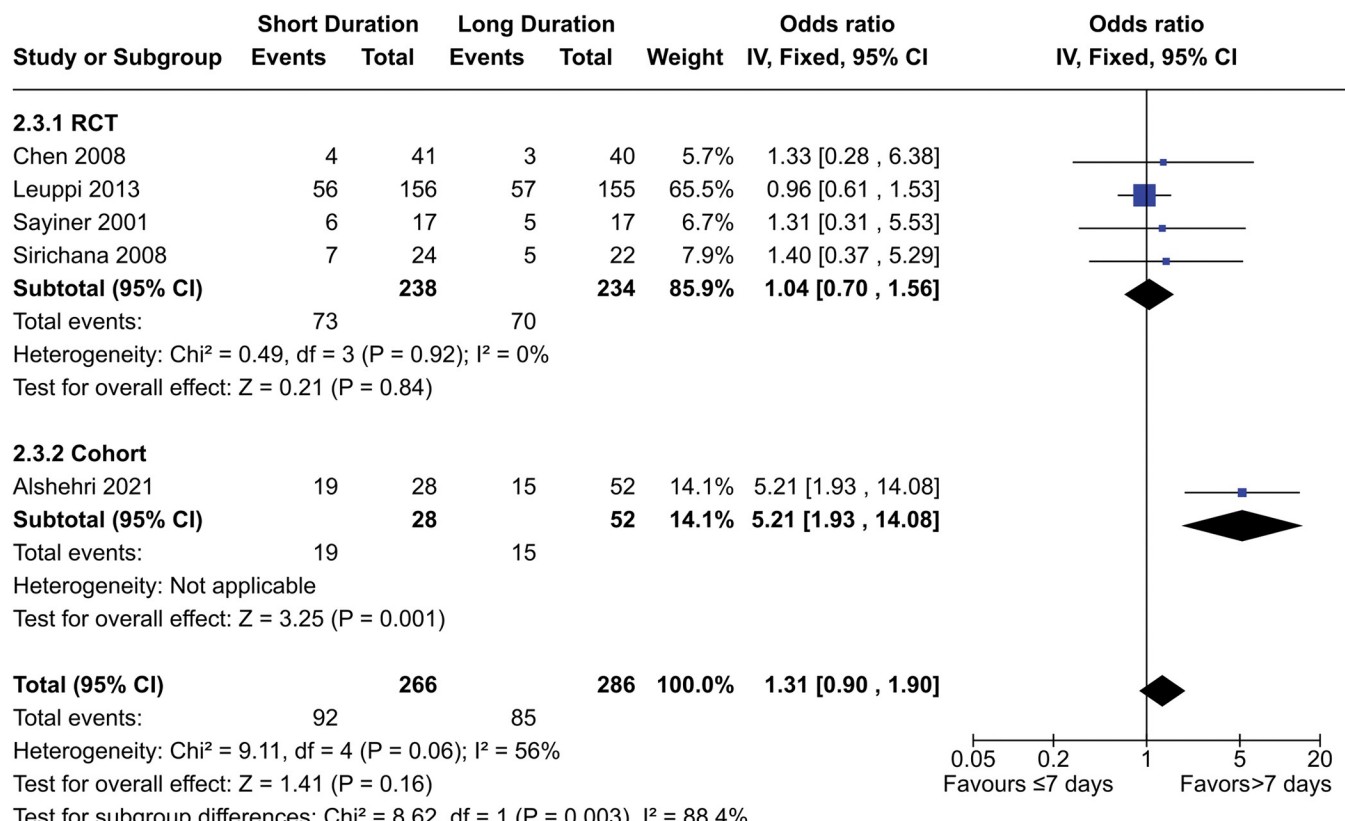

**Fig 4. Comparison of re-exacerbation risk between short- (≤7 days) and long- (>7 days) duration regimens of corticosteroids.**

Sirchana 2008 using a 30-day follow-up while Leuppi 2013 and Sayiner 2001 both using 180 day follow-ups (the follow-up duration for Chen 2008 was unknown). However, data for Leuppi 2013 was non-significant at 30 days via log-rank test (p = 0.87). Sensitivity analysis excluding Alshehri 2021 and Sirichana 2008 does not produce a significant result (OR 1.01, 95% CI = 0.66–1.55, p = 0.95), and neither does changing to a random-effects model (OR 1.62, 95% CI = 0.79–3.32, p = 0.18).

Additionally, Zhou 2021 reported frequency exacerbations as a rate, which was not significant between the two groups (p>0.05) after 1, 3, 6, 9, and 12 months.

Three RCTs (Chen 2008, Leuppi 2013, and Zhou 2021) were deemed to be at low risk of bias, while one (Sirichana 2008) had unknown risk of bias but was unblinded. Zhou 2021 was deemed to be at low risk despite being open-label due to the objectivity of the outcome. Alshehri 2021 had a serious risk of bias. The certainty of evidence was downgraded to "moderate" due to imprecision, while the certainty for the cohort data was downgraded to "very low" due to risk of bias and imprecision. Altogether, our data shows the number of re-exacerbations during follow-up is likely to be either the same between short- (≤7 days) and long- (>7 days) duration regimens or slightly higher in short-duration regimens.

**Hyperglycemia.** Two RCTs and one cohort study contributed a total of 423 participants to the meta-analysis for hyperglycemia (Fig 5). The OR for the combined data was 0.96 (95% CI = 0.59–1.55, p = 0.58), while the OR for the RCTs only was 0.99 (95% CI = 0.64–1.53, p = 0.96). Both RCTs were deemed to be at low risk of bias, while the cohort study had significant bias.

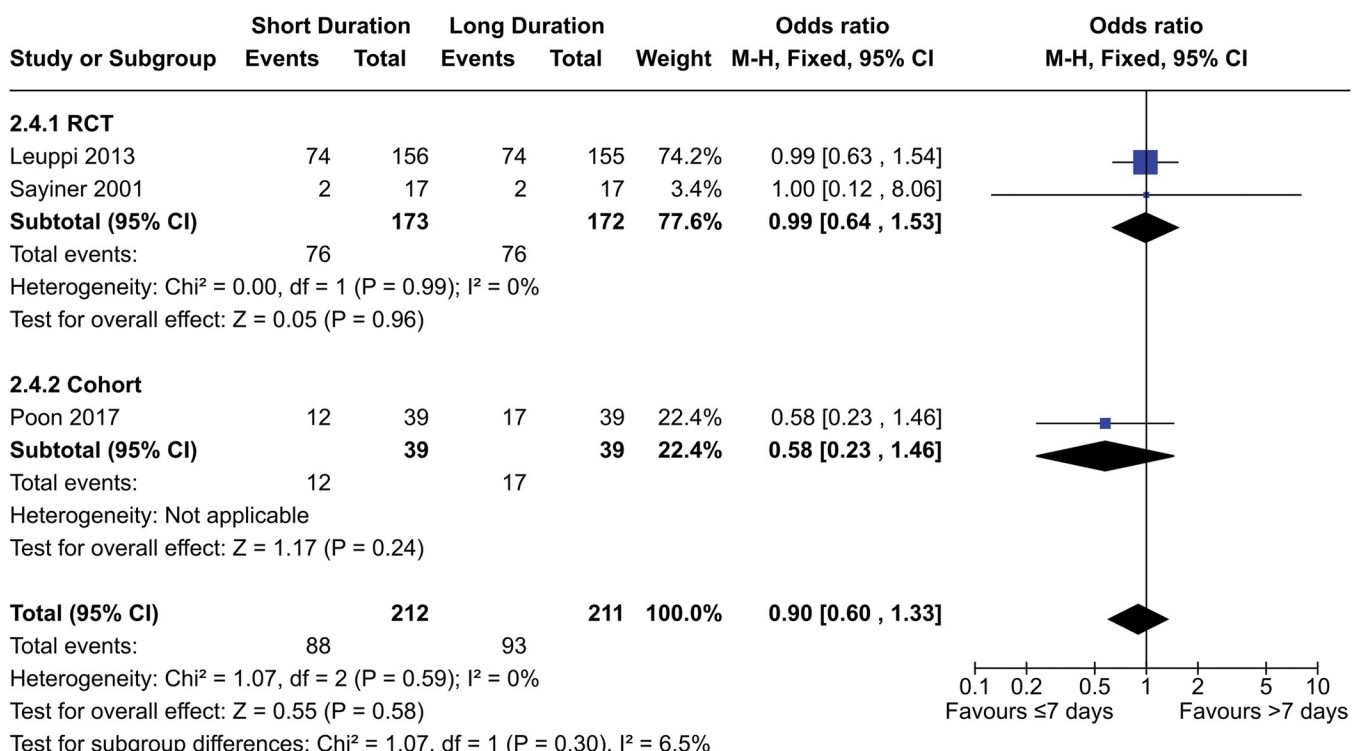

**Fig 5. Comparison of hyperglycemia risk between short- (≤7 days) and long- (>7 days) duration regimens of corticosteroids.**

Additionally, one abstract (Sirichana 2008) noted that fasting plasma glucose was not different after 14 days, though that is only a proxy for hyperglycemia. Also, the study was unblinded and not enough information was available to assess overall risk of bias.

The certainty of evidence from the RCTs was downgraded to "moderate" due to imprecision, and the certainty from the cohort study was downgraded to "very low" due to risk of bias and imprecision. Overall, our data shows there is likely no difference in hyperglycemia incidents between short- (≤7 days) and long- (>7 days) duration regimens.

**Infection incidence.** One RCT and one cohort study contributed 389 patients to the meta-analysis for infection incidence (Fig 6). The OR for the combined data was 0.96 (95% CI = 0.59–1.56, p = 0.87), while the OR for the RCT only was 0.99 (95% CI = 0.61–1.62, p = 0.97).

However, another cohort study (Sivapalan 2019) reported an adjusted hazard ratio of 1.2 (95% CI = 1.0–1.3, p = 0.011) after 1 year, with more incidents in the long-duration group. However, this data only included pneumonia events and only included outpatients, who had comparatively less severe exacerbations.

Leuppi 2013, Sivapalan 2019, and Poon 2020 had low, moderate, and serious risk of bias, respectively. The certainty of evidence from Leuppi 2013 was rated as "moderate" due to imprecision, while the evidence from the cohort studies was rated as "very low" due to risk of bias, imprecision, and indirectness in the case of Sivapalan 2019. Our data shows there is likely to be no difference in infection risk between short- (≤7 days) and long- (>7 days) duration regimens, but there may be less incidents in short-duration regimens specifically in outpatients with less-severe exacerbations.

**$FEV_1$ change.** Three RCTs contributed 161 patients to the meta-analysis of $FEV_1$ change (Fig 7), which used the post-treatment $FEV_1$ (measured on the last day of the long-treatment

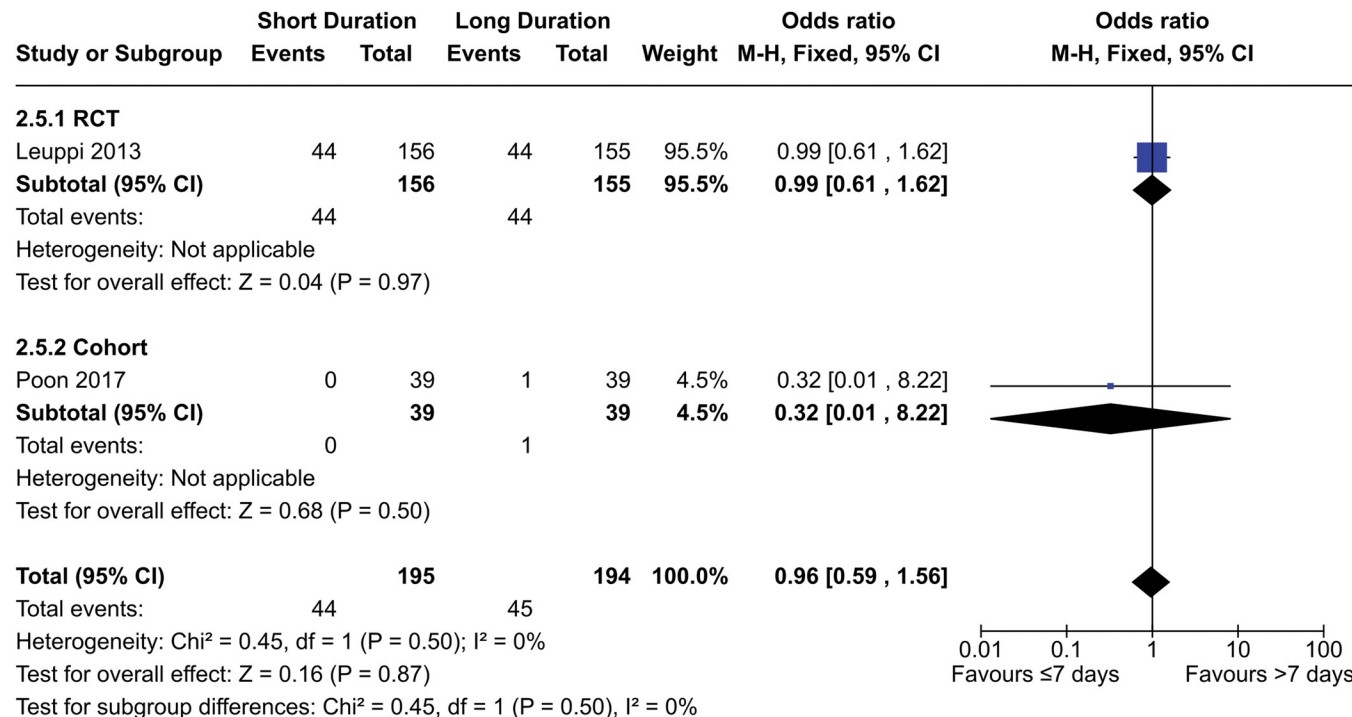

**Fig 6. Comparison of hyperglycemia risk between short- (≤7 days) and long- (>7 days) duration regimens of corticosteroids.**

regimen) as a proxy for the increase in $FEV_1$. This choice was made to increase the number of studies that could be included in the analysis, and because the baseline pretreatment $FEV_1$ between the two groups in each study were not significant. The mean difference was -18.40 mL (95% CI = -111.80–75.01 mL, p = 0.70), but there was considerable heterogeneity in the studies ($I^2$ = 71%). Compared to the other two studies, data from Sayiner 2001 had a mean difference that favoured long duration regimens. This may be due to the relatively short duration of treatment of 3 days for the experimental arm, while the short duration arm for Chen 2008 and Sirichana 2008 were 7 and 5 days, respectively. If this is the case, this heterogeneity may represent a form of dose-dependent relationship between corticosteroid regimen duration and $FEV_1$ change. Previous research suggests effect of corticosteroids on $FEV_1$ recovery may be most apparent in the initial 3–5 days of treatment [42]. In that case, the short-duration group of Sayiner 2001 would have ended at the beginning of this period. Neither excluding the study nor switching to a random-effects model produced a statistically significant result (MD 54.50

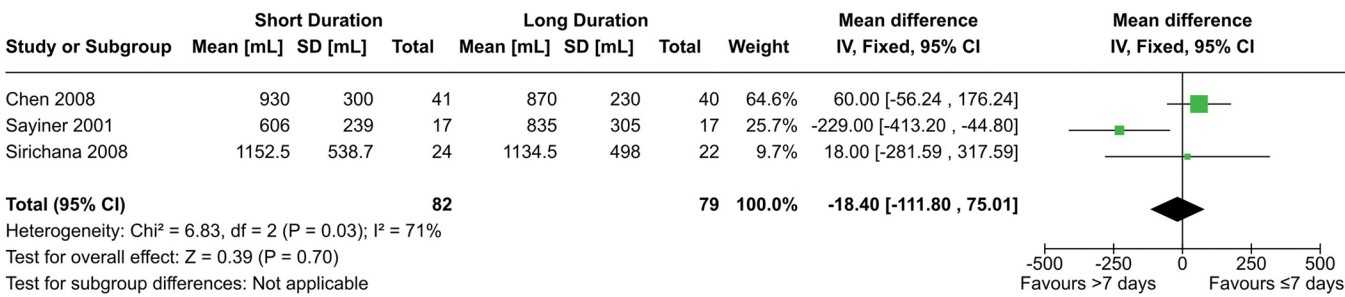

**Fig 7. Comparison of $FEV_1$ change between short- (≤7 days) and long- (>7 days) duration regimens of corticosteroids.**

mL, 95% CI = -53.87–162.88 mL, p = 0.32, and MD -49.90 mL, 95% CI = -251.05–151.26 mL, p = 0.63, respectively). Excluding Sirichana 2008 with or without excluding Sayiner 2001 also did not produce a significant result (MD 60 mL, 95% CI = -56.24–176.24 mL, p = 0.66 and MD -22.32 mL, 95% CI = -120.62–75.99 mL, p = 0.31, respectively).

Leuppi 2013 found no difference in $FEV_1$ at discharge, nor at 0, 6, 30, or 180 days (p = 0.94). Gomaa 2008 also reported no significant difference in $FEV_1$ change after 7, 14, and 30 days. Al Mamun 2011 found no significant difference at days 7 and 14 (p = 0.100, p = 0.079, respectively). Zhou 2021 only provided data in terms of % of predicted $FEV_1$. Neither the % predicted baseline $FEV_1$ nor % predicted $FEV_1$ after 180 days were significant across the two groups (p = 0.174 and p = 0.134 respectively), with the mean being slightly higher in the long duration arm in both time points.

Leuppi 2013, Chen 2008, and Sayiner 2001 had low risk of bias. Sirichana 2008 and Zhou 2021 were both open-label, with the former having unknown overall risk of bias and the latter having moderate risk of bias for this outcome. Gomaa 2008 and Al Mamun 2011 had an unknown risk of bias. The certainty of evidence for this outcome was downgraded to "low" due to imprecision in the effect estimate and inconsistency from unexplained heterogeneity. Our data shows there may be no difference in $FEV_1$ change between short- (≤7 days) and long- (>7 days) duration regimens.

## Discussion

To our knowledge, this review addresses the question of optimal corticosteroid regimen duration in COPD exacerbations using the largest dataset to date. The impetus of this study was the conclusion of the 2018 Cochrane review by Walters et al., which concluded that although 7- and 14-day regimens did not seem result in significantly different outcomes, there was a relative paucity of evidence that necessitated further research [28]. In our review, we included additional data from a large recently published RCT (Zhou 2021), three retrospective cohort studies, and qualitative data which could not be included in the meta-analyses. Overall, we believe this new data reinforces the conclusion that short-duration regimens are not worse than long-duration regimens in terms of mortality, re-exacerbation, hyperglycemia, hospital LOS, and $FEV_1$ change. Notably, Sivapalan 2019 was a relatively large and rigorous cohort study which specifically examined outpatient data unlike the other studies which primarily focused on hospitalized patients. It reported clinically relevant decreases in mortality and pneumonia incidents in the short-duration regimen. While it is only a single study examining two outcomes, it validates previous concerns regarding the non-generalizability of research from severe exacerbations to those with milder exacerbations [28]. This is also consistent with the GOLD guidelines, which do not recommend the use of corticosteroids in treating mild exacerbations [9]. The European Respiratory Society and American Thoracic Society tentatively suggest consideration of 14 days or fewer of oral corticosteroid use in ambulatory patients with exacerbations [14]; the findings of Sivapalan 2019 suggest that duration should be decreased even further.

### Limitations of our study

Although we were able to include more studies compared to past reviews, our analyses still suffered from a relative lack of data. Of the 11 studies we included in our review, only 4 of them were full-article RCTs, and only 2 of those RCTs had more than 100 participants. This issue manifested as wider CIs in the summary estimates, which resulted in us downgrading the certainty of every outcome due to imprecision since we felt that the clinical decisions would be different at either end of the CI. Additionally, we were hesitant to give much weight to the

cohort studies during our interpretation of the results due to their small size and inherent risk of bias [43]. The low number of studies also meant we were unable to perform subgroup analysis, so the effects of prognostic factors such as blood eosinophil count and age were unable to be assessed [44]. As $FEV_1$/FVC and % predicted $FEV_1$ tends to misdiagnose very young and old patients, this may have impacted the results of the studies used in this review [12, 14]. In our review, several of the abstract-only studies did not specify a minimum age for the participants, while Poon 2020 included all patients over 18 years old. However, the mean/median ages of the studies were generally between 60 and 70 years, and given that COPD mostly affects older patients, we do not believe the inclusion of younger patients will significantly affect the overall conclusions. Despite this, it would have been interesting to do a subgroup analysis by age, or even reclassify each study's participants using z-scores, but this was not possible. While all the large RCTs explicitly used the GOLD criteria to define COPD and exacerbations, some of the other studies—especially abstracts—did not specify any. In particular, Sayiner 2001 and Wood-Baker 1997 were conducted before the GOLD guidelines were established, increasing the risk that their definitions were different from those we use today. However, their data is not heterogenous with the rest of the studies, so we do not have clear reason to believe that is the case.

There was heterogeneity in the specifics of the corticosteroid regimens used in each study in terms of corticosteroid type, administration methods, treatment duration/dosage, and the presence of a taper. For example, while Wood-Baker 1997 compared 3- and 14-day regimens, the cumulative dosages were much more similar at 7.5 mg/kg and 6.3 mg/kg respectively [41]. The short-duration regimens of the included studies ranged from 3 to 7 days, while the long-duration regimens ranged from greater than 5 days to over 16.5 days. However, data between trials using different regimen lengths were generally not heterogenous with one another, suggesting that the clinical effects of corticosteroids between each of those ranges are minimal. The main exception is Sayiner 2001's $FEV_1$ data, which as stated in the results section may be due to the duration being genuinely lower than the physiologically optimal duration [42]. However, this is only one relatively small and old RCT, so whether this is the case is unclear. There was also heterogeneity in the settings of each study. For example, Alshehri 2021 was conducted in Saudi Arabia and reported significantly more re-exacerbations in the short-duration group, contrary to other studies [36]. The authors cited differences in treatment adherence as a possible explanation; another study found that patients in Saudi Arabia had lower adherence to COPD maintenance treatment compared to those in Turkey—where Sayiner 2001 was conducted [45]. Poor adherence to guidelines during exacerbations remains an issue and is related to worse outcomes for patients [26, 27]. This reiterates the concern that data from our included studies may not be generalizable to all populations. Relatedly, our sole study that specifically assessed outpatients raises questions about the generalizability of our results in outpatients, which represent over 80% of all exacerbations [9]. RECUT—a currently ongoing RCT which specifically recruits outpatients—will hopefully shed more light on this issue [42].

While we adhered to the PRISMA guidelines and other best practices whenever possible, some limitations exist in the methods we used. We were unable to statistically or graphically check for publication bias due to an inadequate number of studies. Such tests generally require at least 10 studies to be sufficiently powered [46], while the most studies we had in an outcome was five. None of the full articles reported conflicts of interests arising from funding or other sources (abstracts were unable to be assessed) or were designed/reported in an obviously intentionally biased way. We do not believe considerable publication bias was present, but we cannot be certain. Another limitation is that we did not report on all outcomes that were relevant to the topic. We decided on our outcomes *a priori* to minimize our ability to perform

selective reporting. Some outcomes we did not address were mean time to re-exacerbation, treatment failure, quality of life, other spirometric parameters such as FVC, and other adverse effects. A past review by Walters et al. did not find any significant difference in any of those outcomes [28], although notably, Zhou 2021 found a significant decrease in the mean time to next exacerbation in the short-duration cohort [17].

## Implications for the future

The certainties of evidence for each outcome were all either "moderate" or "low" for RCT data, and "very low" for cohort study data. The current body of evidence is heterogenous and too spare for more sophisticated analysis. While short-duration regimens are unlikely to be worse than long-duration regimens in general, we cannot fully preclude the possibility of inferiority for all individual outcomes, especially re-exacerbation rate. Additional research is needed, with data on hospital LOS, re-exacerbation, and $FEV_1$ change being the most likely to change the summary estimate in a clinically relevant way. Additionally, preliminary data suggests short-duration regimens may be considerably preferable to long-duration regimens for patients with milder exacerbations, but further data from RCTs are needed to confirm this. Overall, our findings support the recommendations of GOLD that systemic corticosteroids are given for 5–7 days for moderate to severe COPD exacerbations [9].

## Supporting information

**S1 Fig. Completed PRISMA 2020 checklist.**
(PDF)

**S2 Fig. MEDLINE search strategy example.**
(PDF)

**S1 Table. List of excluded studies and reasons for exclusion.**
(PDF)

## Acknowledgments

We would like to thank Winston Hou for providing guidance throughout this study.

## Author Contributions

**Conceptualization:** Zhen Zhao, Wei Ting Xiong.

**Data curation:** Zhen Zhao.

**Formal analysis:** Zhen Zhao, Owen Lou, Yiyang Wang, Raymond Yin, Florence Deng, Jing Yi Xie.

**Investigation:** Zhen Zhao, Owen Lou, Yiyang Wang, Raymond Yin, Ethan C. Wu, Jing Yi Xie, Jerry Wu, Avery Ma.

**Methodology:** Owen Lou, Yiyang Wang, Raymond Yin.

**Project administration:** Zhen Zhao, Wei Ting Xiong.

**Resources:** Zhen Zhao.

**Supervision:** Zhen Zhao, Wei Ting Xiong.

**Validation:** Zhen Zhao, Owen Lou, Avery Ma.

**Visualization:** Zhen Zhao.

**Writing – original draft:** Zhen Zhao, Yiyang Wang, Raymond Yin, Ethan C. Wu, Jing Yi Xie, Yongzhi Guo.

**Writing – review & editing:** Zhen Zhao, Carrie Gong, Florence Deng, Jing Yi Xie, Jerry Wu, Avery Ma.

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
