## [Decision Letter · Decision Letter 0]

24 Oct 2023

PONE-D-23-26586Long- versus short-duration systemic corticosteroid regimens for acute exacerbations of COPD: A systematic review and meta-analysis of randomized trials and cohort studiesPLOS ONE

Dear Dr. Zhao,

Thank you for submitting your manuscript to PLOS ONE. After careful consideration, we feel that it has merit but does not fully meet PLOS ONE’s publication criteria as it currently stands. Therefore, we invite you to submit a revised version of the manuscript that addresses the points raised during the review process.

Please consider all reviewer comments. 

We look forward to receiving your revised manuscript.

Kind regards,

Tim Mathes

Academic Editor

PLOS ONE

Journal Requirements:

Reviewers' comments:

Reviewer's Responses to Questions

**Comments to the Author**

1. Is the manuscript technically sound, and do the data support the conclusions?

Reviewer #1: No

Reviewer #2: Partly

Reviewer #3: Yes

2. Has the statistical analysis been performed appropriately and rigorously? 

Reviewer #1: Yes

Reviewer #2: Yes

Reviewer #3: I Don't Know

3. Have the authors made all data underlying the findings in their manuscript fully available?

Reviewer #1: Yes

Reviewer #2: Yes

Reviewer #3: Yes

4. Is the manuscript presented in an intelligible fashion and written in standard English?

Reviewer #1: No

Reviewer #2: Yes

Reviewer #3: Yes

5. Review Comments to the Author

Reviewer #1: Abstract::Please tell the number of includrd studies. Number of patients is not enough.

OR mean nothing without telling in favor of which study arme

Introduction : should focus on the objective. You are not dealing with COPD or COPD exacerbation. Rather the topic is systemic corticosteroid duration.

What conclusion would you draw from studies comparing three or more days and those comparing 5 or more days ? when you set a limit the duration to 7 days.

Reviewer #2: Major aspect: The latest review about this topic was published in 2018. Was this updated systematic review necessary? The motivation behind the new systematic review remains unclear. You should explicitly explain the reasons why you did in the introduction section more clearly and summarize the first part of the Introduction section (too long).

Abstract: When describing odds ratios for different outcomes, it is essential to specify the treatment arm. In the first comparison, clarify whether you are referring to short-term or long-term corticosteroid use. Additionally, provide the definition of the acronym MD the first time it is mentioned.

Delete the sentence at the end of the Introduction section ‘However, there was still not enough data for a conclusive verdict, and further studies would be beneficial.’. This belongs to discussion.

Regarding the two reviewers who independently assessed each study, ensure compliance with PRISMA requirements by specifying their identities.

Why did you consider studies including adults older than 18 years if you were assessing the benefit of corticosteroids in COPD exacerbations. You should have limited the population included to adults aged 40 or more. Some studies included have been published some decades ago and the risk of having included diagnoses other than COPD is not negligible. Please explain.

In connection with the previous question, clarify whether only patients with spirometrically-diagnosed COPD were included. This information is not clearly described in Table 1.

The references section is missing some critical information. One example of this is the improper referencing of the Walters review.

Reviewer #3: 10/22/2023

Review of manuscript entitled "Long- versus short-duration systemic corticosteroid regimens for acute exacerbations of COPD: A systematic review and meta-analysis of randomized trials and cohort studies".

The authors conducted a systematic review on the issue of duration of systemic corticosteroid use during a COPD exacerbation, looking at various outcome measures, including mortality and adverse events, and adding recent data to an existing review.

I would like to congratulate all the authors on the manuscript as it is very well written and shows great competence in the field of systematic literature review and meta-analysis.

My comments can be found below:

Minor comments:

1. In the introduction I would suggest describing alternative criteria for diagnosis as well as severity, e.g. by classifcation using Z-scores and the lower limit of normal of pulmonary function, as this is supported by recent ERS/ATS guidelines that contradict GOLD.

2. I would suggest including in the introduction an explanation of the aetiology of COPD, particularly tobacco abuse.

3. Please explain what you mean by "a third common factor" in relation to COPD exacerbation.

4. Please specify the adverse effects of systemic corticosteroids during the relatively short term use (14 days) and discuss clinical relevancy of discrimination between long and short term application with regard to adverse effects

Major comments:

1. Please explain how you selected your outcome parameters a priori and whether you consulted a clinical expert for clinical relevance and whether an applicability analysis was performed in this context.

Overall, the authors adequately addressed the limitations of the study and presented the results in accordance with these limitations.

6. PLOS authors have the option to publish the peer review history of their article (what does this mean?). If published, this will include your full peer review and any attached files.

Reviewer #1: No

Reviewer #2: **Yes: **Carl Llor

Reviewer #3: No

---

## [Author Response · Author response to Decision Letter 0]

7 Dec 2023

To the editor and peer reviewers,

 On behalf of all the authors, I would like to thank you for your feedback, it has been tremendously helpful to us. We have revised our manuscript and have provided a response to each of your points below. We hope that our revisions resolve the comments and concerns you have brought up, and we would gladly incorporate any further feedback you may have.

Sincerely,

Zhen Zhao

Reviewer 1

Abstract:

Please tell the number of includrd studies. Number of patients is not enough.

OR mean nothing without telling in favor of which study arme

- We have adjusted the abstract to include this information.

Introduction

should focus on the objective. You are not dealing with COPD or COPD exacerbation. Rather the topic is

systemic corticosteroid duration

- We have edited our introduction to place less emphasis on COPD and exacerbations, decreasing the amount of less-relevant information. In response to another reviewer’s feedback, we have included a section about z-scores and the LLN as alternatives to GOLD’s definition of COPD. We feel it illustrates the fact that there is still active research in the treatment of COPD, and how alternative criteria can affect the results of clinical trials.

What conclusion would you draw from studies comparing three or more days and those comparing 5 or more days ? when you set a limit the duration to 7 days.

- This is a source of methodological heterogeneity in our data, which can be a confounding factor. Interestingly though, the data from the studies with shorter durations were not statistically heterogenous with, suggesting that this variation in the durations might not be clinically significant. The same goes for the different durations in the long-duration arms. The exception to this is Sayiner 2001’s FEV1 change data, which was quite different from the others. One explanation is that the 3-day regimen is shorter than the optimal duration. The RECUT trial article (Urwyler 2019) says that evidence suggests the greatest FEV1 increase occurs at 3-5 days, so if that’s true then Sayiner 2001’s 3-day duration would be too short. We discuss this in the results section for FEV1. 

Reviewer 2

The latest review about this topic was published in 2018. Was this updated systematic review necessary? The motivation behind the new systematic review remains unclear. You should explicitly explain the reasons why you did in the introduction section more clearly and summarize the first part of the Introduction section (too long).

- We summarized the first parts of the introduction, placing less emphasis on COPD and more on the use of corticosteroids in exacerbations itself. We conducted this review because the previous one by Walters et al. found that there was not enough data to make confident conclusions about regimen durations. Our review includes additional studies which increase the confidence in the conclusions, including the first study to study patients with milder COPD. We stated this at the end of our introduction.

Abstract: When describing odds ratios for different outcomes, it is essential to specify the treatment arm. In the first comparison, clarify whether you are referring to short-term or long-term corticosteroid use. Additionally, provide the definition of the acronym MD the first time it is mentioned.

- We adjusted the abstract’s wording to explicitly state the short duration is the treatment arm. The acronym MD is actually defined earlier in the abstract along with OR, but we have modified the wording to hopefully make this more apparent.

Delete the sentence at the end of the Introduction section ‘However, there was still not enough data for a conclusive verdict, and further studies would be beneficial.’. This belongs to discussion.

- We removed this sentence from the introduction.

Regarding the two reviewers who independently assessed each study, ensure compliance with PRISMA requirements by specifying their identities.

- We have added this to the end of the Data Extraction section in the methods. We were unsure if you wanted us to specifically state which authors assessed each individual article or just which authors did assessments at all, and the PRISMA 2020 checklist is ambiguous, but we have done the former in our revision.

Why did you consider studies including adults older than 18 years if you were assessing the benefit of corticosteroids in COPD exacerbations. You should have limited the population included to adults aged 40 or more. Some studies included have been published some decades ago and the risk of having included diagnoses other than COPD is not negligible. Please explain.

- While it’s true that COPD mostly affects older people, it can still affect those younger than 40 years old. Several abstracts we used did not state whether they only included patients greater than 40, while one of the retrospective studies (Poon 2017) included all patients older than 18. As the included patients for each study should still mostly be relatively old, we don’t expect the few young patients to significantly affect the results of the study and chose to use a relatively liberal criteria in terms of age to avoid excluding studies. However, we had added a part to the discussion addressing this concern.

- We’ll address the point regarding misdiagnoses along with your next comment.

In connection with the previous question, clarify whether only patients with spirometrically-diagnosed COPD were included. This information is not clearly described in Table 1.

- We’ve updated Table 1 to make it clear if studies are using the spirometric GOLD criteria for COPD. While all the large studies use the GOLD criteria, some other studies do not explicitly specify what their criteria for COPD and/or exacerbations. However, we have chosen to include them because it is likely that they are indeed using the GOLD definitions, with most of them discussing the GOLD guidelines in the text. Sayiner 2001 and Wood-Baker 1997 were conducted before GOLD reports began being made, so there is an increased risk that their definitions of COPD would be different from today, like you mentioned. However, previous systematic reviews like the Walters 2018 have also included them, and we don’t have a clear reason to suspect that they are using a drastically different definition. Their data is not different from the other studies to a point that suggests considerable heterogeneity is present. So while we will keep them in our analyses, we have also added a section to the discussion talking about these concerns.

The references section is missing some critical information. One example of this is the improper referencing of the Walters review.

- We looked over the references section and fixed several mistakes. For the Walters reviews, we noticed not all of the initials for the authors was included; we hope that addresses the issues you observed.

Reviewer 3

1. In the introduction I would suggest describing alternative criteria for diagnosis as well as severity, e.g. by classification using Z-scores and the lower limit of normal of pulmonary function, as this is supported by recent ERS/ATS guidelines that contradict GOLD.

- We added a section describing z-scores and LLNs, and the benefits/disadvantages of them compared to the ratio-based criteria GOLD recommends. 

2. I would suggest including in the introduction an explanation of the aetiology of COPD, particularly tobacco abuse.

- We have added a sentence about tobacco and other aetiological factors. However, in response to other feedback, we are trying to focus less on COPD and more specifically on exacerbations and corticosteroid treatment itself, so we have not chosen to discuss the aetiology in detail.

3. Please explain what you mean by "a third common factor" in relation to COPD exacerbation.

- We meant to say it is unknown if exacerbations had a causal effect on worsening COPD, or if there is a third “common” confounding factor. We’ve changed our wording to make it clearer. Note that we have changed the location of this sentence to the end of the first paragraph.

4. Please specify the adverse effects of systemic corticosteroids during the relatively short term use (14 days) and discuss clinical relevancy of discrimination between long and short term application with regard to adverse effects

- We already had a sentence describing the adverse effects of corticosteroids in general, but we’ve changed it to specifically specify the effects during short-term use. In terms of clinical relevancy of discrimination between regiment durations, we explain that the severity of adverse effects is correlated with the dose and duration of corticosteroid usage. Also, we added a sentence noting how the recommended duration has decreased over the recent decades, showing that there is a real interest to minimize the duration of corticosteroid regimens.

5. Please explain how you selected your outcome parameters a priori and whether you consulted a clinical expert for clinical relevance and whether an applicability analysis was performed in this context.

- To pick our outcomes, we reviewed the current literature, paying particular attention to outcomes used in previous systematic reviews and trials on corticosteroids and COPD. We consulted a physician to make sure our outcomes were relevant, but we did not do a formal applicability analysis. We believe our outcomes are relevant, as they are often used in a clinical context, frequently referenced in the literature, and often appear in other clinical trials studying corticosteroids.

---

## [Decision Letter · Decision Letter 1]

14 Dec 2023

Long- versus short-duration systemic corticosteroid regimens for acute exacerbations of COPD: A systematic review and meta-analysis of randomized trials and cohort studies

PONE-D-23-26586R1

Dear Dr. Zhao,

We’re pleased to inform you that your manuscript has been judged scientifically suitable for publication and will be formally accepted for publication once it meets all outstanding technical requirements.

Kind regards,

Tim Mathes

Academic Editor

PLOS ONE

Additional Editor Comments (optional):

Reviewers' comments:

Reviewer's Responses to Questions

**Comments to the Author**

1. If the authors have adequately addressed your comments raised in a previous round of review and you feel that this manuscript is now acceptable for publication, you may indicate that here to bypass the “Comments to the Author” section, enter your conflict of interest statement in the “Confidential to Editor” section, and submit your "Accept" recommendation.

Reviewer #2: All comments have been addressed

2. Is the manuscript technically sound, and do the data support the conclusions?

Reviewer #2: Yes

3. Has the statistical analysis been performed appropriately and rigorously? 

Reviewer #2: Yes

4. Have the authors made all data underlying the findings in their manuscript fully available?

Reviewer #2: Yes

5. Is the manuscript presented in an intelligible fashion and written in standard English?

Reviewer #2: Yes

6. Review Comments to the Author

Reviewer #2: The paper reads better. Some inquiries persist due to the quality of the papers incorporated in this systematic review. However, it's important to note that this limitation is inherent to studies of this nature and you have appropriately addressed this.

7. PLOS authors have the option to publish the peer review history of their article (what does this mean?). If published, this will include your full peer review and any attached files.

Reviewer #2: **Yes: **Carl Llor

---

## [Editor Report · Acceptance letter]

18 Dec 2023

PONE-D-23-26586R1 

PLOS ONE

Dear Dr. Zhao, 

I'm pleased to inform you that your manuscript has been deemed suitable for publication in PLOS ONE. Congratulations! Your manuscript is now being handed over to our production team.

Kind regards, 

on behalf of

Dr. Tim Mathes 

Academic Editor

PLOS ONE